theoretical biology, health and disease and epidemiology, ecology

COVID-19, critical slowing down, early warning signals, timescale separation

**Author for correspondence:**
Fabian Dablander
e-mail: dablander.fabian@gmail.com

# Overlapping timescales obscure early warning signals of the second COVID-19 wave

Fabian Dablander[1], Hans Heesterbeek[2], Denny Borsboom[1] and John M. Drake[3,4]

[1]Department of Psychological Methods, University of Amsterdam, Amsterdam, The Netherlands
[2]Department of Population Health Sciences, Utrecht University, Utrecht, The Netherlands
[3]Odum School of Ecology, and [4]Center for the Ecology of Infectious Diseases, University of Georgia, Athens, GA, USA

FD, 0000-0003-2650-6491; JMD, 0000-0003-4646-1235

Early warning indicators based on critical slowing down have been suggested as a model-independent and low-cost tool to anticipate the (re)emergence of infectious diseases. We studied whether such indicators could reliably have anticipated the second COVID-19 wave in European countries. Contrary to theoretical predictions, we found that characteristic early warning indicators generally *decreased* rather than *increased* prior to the second wave. A model explains this unexpected finding as a result of transient dynamics and the multiple timescales of relaxation during a non-stationary epidemic. Particularly, if an epidemic that seems initially contained after a first wave does not fully settle to its new quasi-equilibrium prior to changing circumstances or conditions that force a second wave, then indicators will show a decreasing rather than an increasing trend as a result of the persistent transient trajectory of the first wave. Our simulations show that this lack of timescale separation was to be expected during the second European epidemic wave of COVID-19. Overall, our results emphasize that the theory of critical slowing down applies only when the external forcing of the system across a critical point is slow relative to the internal system dynamics.

## 1. Introduction

Forecasting the (re)emergence of infectious diseases is of great importance to public health [1–6]. In recent years, early warning indicators based on the phenomenon of critical slowing down have been suggested as a way to anticipate transitions in a wide range of dynamical systems (for overviews, see e.g. [7–12]). Critical slowing down describes the phenomenon that many systems, as they approach their critical point, return more slowly to their equilibrium after small external perturbations, resulting in an increase in statistics such as the local autocorrelation coefficient and variance [13,14]. In standard models of infectious disease transmission, major outbreaks are possible when the effective reproductive number, $R_t$, is greater than one. The threshold $R_t = 1$ corresponds to a (dynamic) transcritical bifurcation, which is a type of bifurcation that is preceded by critical slowing down [15,16]. Early warning indicators based on critical slowing down have been studied extensively and led to a promising research line that aims to utilize them as a tool to forecast the (re)emergence as well as the elimination of infectious diseases (e.g. [7,17–28]).

In light of this prior research, it seems natural to ask whether early warning indicators based on critical slowing down could have allowed us to anticipate the second COVID-19 wave (e.g. [29,30]) and if not, how this can be understood. Here, we question the applicability of early warning indicators in the COVID-19 context, because the COVID-19 situation violates a key assumption of the theory of critical slowing down: a separation of timescales such that the dynamics of the epidemic settle down to a quasi-equilibrium from which there is a slow drift towards

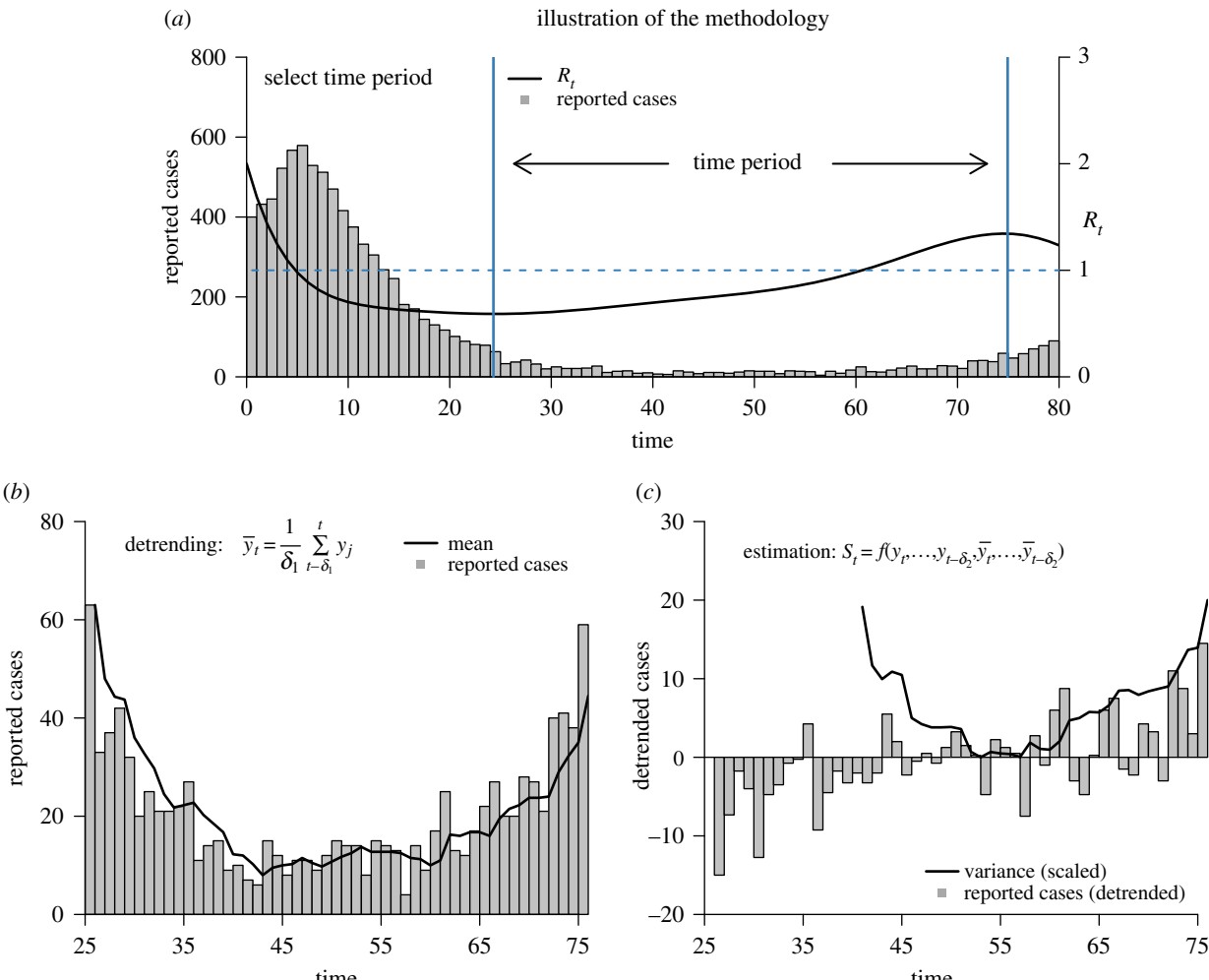

**Figure 1.** Illustration of our methodology on simulated data. Panel (*a*) shows reported cases (grey) and $R_t$ (black). Vertical blue lines indicate the minimum and maximum $R_t$ after the first wave receded. Panel (*b*) shows reported cases (grey) during the selected time period and an estimate of the mean (black) using a rolling window of size $\delta_1 = 4$. Panel (*c*) shows detrended cases (grey) and an estimate of the (scaled) variance (black) using a rolling window of size $\delta_2 = 15$. (Online version in colour.)

the critical point. The quasi-equilibrium corresponds to the dynamics of the epidemic being subcritical ($R_t < 1$) but not dying out due to the importation of cases, instead reaching a quasi-stationary state. To our knowledge, there is presently no theory that would indicate whether early warning signals, under such commensurate timescales, can be expected to be reliable. In this paper, we report on a combination of empirical analysis and simulation studies to investigate this issue. Focusing on Europe, we find that a suite of early warning indicators did not reliably rise prior to the second wave in any country as the classical theory of critical slowing down would predict. Using a simulation study that mimics the COVID-19 situation—a first outbreak closely followed by a second one—we show that this contradictory result can be fully explained by the fact that, in the case of COVID-19, in almost all countries $R_t$ already began to creep up again before the number of case reports stabilized at a low value after the first wave. These results indicate that caution is warranted in applying early warning indicators to highly non-stationary settings, such as multi-wave epidemics.

## 2. Early warning signals for COVID-19 in Europe

In this section, we quantify the extent to which early warning indicators increased prior to the second wave in a number of European countries.[1] We outline our methodology aided by figure 1 in §2(a), and report our results in §2(b).

### (a) Methods
#### (i) Estimation of $R_t$
To identify the time at which the COVID-19 epidemic became supercritical for the second time in each country, we followed Gostic *et al.* [31] to estimate the instantaneous $R_t$ using the method of Abbott *et al.* [32], which improves upon Cori *et al.* [33]. The method simultaneously estimates the incidence of infections and $R_t$ using Bayesian latent variable modelling. The method proceeds in two steps. First, the incidence at each time step is estimated by convolving the previous number of infections with a probability distribution for the generation interval. This incidence is then convolved over an uncertain incubation period and reporting delay distribution to yield the reported cases (for details, see Abbott *et al.* [32]). We applied this method to a broad range of European countries using daily case report data from the WHO, spanning the period from March to October 2020. We used the R package *covidregionaldata* to load the raw data [34]; no further preprocessing was necessary.

#### (ii) Selecting the time period between waves
Next, we selected a time period in which to search for evidence of critical slowing down. Early warning indicators are sensitive to changes in the effective reproductive number, and should rise prior to the critical point $R_t = 1$ [7,17]. Using our country-specific estimate for $R_t$, we defined the start and end date of the time series on which we computed the early warning indicators as

follows. We chose as start date the date at which $R_t$ was at its lowest point before reaching $R_t = 1$ prior to the second wave. Similarly, we chose as end date the date at which $R_t$ was at its maximum (before going down again) after it crossed $R_t > 1$. Figure 1a illustrates this selection procedure on a simulated example, with the black line showing $R_t$ and the vertical blue lines indicating its respective minimum and maximum after the first wave receded. We chose this criterion for two reasons. First, after $R_t$ drops below 1, it continues to decrease in all European countries, and we would thus expect early warning indicators to fall, rather than rise. Figure 1a shows a characteristic bifurcation delay (see also §3(a)) that describes that cases lag behind the equilibrium value consistent with $R_t < 1$. Choosing for the starting date the time of the minimum value of $R_t$ before $R_t$ rises again allows the system to come closer to its new equilibrium value. Similarly, we chose to end the interval with the maximum of $R_t$ after it crosses the threshold as a principled approach that could be systematically applied to all data yielding the longest time series.

Figure 2 shows the reported (grey) and estimated true number of cases (black) across European countries, with vertical blue lines indicating the segment of the time series for which we calculated early warning indicators. Figures S1–S5 in the electronic supplementary material, provide a more detailed picture, showing European countries together with their estimated effective reproduction numbers.

### (iii) Detrending and estimation of early warning indicators

As illustrated in figure 1b,c, we detrended the time segment of interest and then estimated early warning indicators using backwards rolling windows with a uniform kernel (i.e. equally weighted past observations) and window sizes $\delta_1$ and $\delta_2$, respectively. A backward rolling window only uses data from the past to estimate the current value of a particular statistic. For example, to estimate the mean at time point $t$, we calculate:

$$\bar{y}_t = \frac{1}{\delta_1} \sum_{j=t-\delta_1}^{t} y_j,$$

where $y_j$ is the number of reported cases at a particular time point $j$ (see black line in figure 1b, for an example). Other early warning indicators we studied were variance, coefficient of variation, index of dispersion, autocovariance, autocorrelation, decay time, skewness, kurtosis and first differenced variance (for mathematical definitions, see [25], table 3).[2] All of these indicators require an estimate of the mean, and so we first estimated the mean and then estimated the particular early warning indicator using a rolling window size of $\delta_2$ (except for the mean, for which we use a window of size $\delta_1$). For example, the variance at time point $t$, which is shown in figure 1c, is calculated as

$$s_t = \frac{1}{\delta_2} \sum_{j=t-\delta_2}^{t} (y_j - \bar{y}_j)^2.$$

We conducted sensitivity analyses with rolling windows of size $\delta_1 \in [2, 4, \ldots, 18, 20]$ for detrending (estimating the mean) and rolling windows of size $\delta_2 \in [5, 6, \ldots, 30]$ for indicator estimation (with the mean being the exception) using the R package spaero [35]. A window size of 10, for example, means that the previous 10 data points are being used to compute the statistic at the current time point. To create a sampling distribution under the null hypothesis of no increase in the early warning indicators that respects the temporal ordering of the data, we fitted a series of ARMA($p, q$) models with $(p, q) \in [1, 2, \ldots 5]$ to the country-specific data. We selected the best-fitting model using AIC and subsequently generated 500 surrogate time series from it, computed the early warning indicators as outlined above, and estimated the rank correlation of the indicator values $s_t$ with time $t$, known as Kendall's $\tau$. This

resulted in the sampling distribution under the null assumption of stationarity, which allowed us to test the actually observed Kendall's $\tau$ against a significance level $\alpha$. This approach is the most widely used approach when estimating and testing early warning indicators using rolling windows [36].

### (b) Results

Figure 3 reports results for European countries for $\delta_1 = 4$ and $\delta_2 = 15$. It shows the value of Kendall's $\tau$ across all early warning indicators, colouring in red the countries for which $\tau$ was either significantly smaller or significantly larger than values generated from the best-fitting country-specific ARMA($p, q$) at $\alpha = 0.05$. Notably, many countries displayed a significant *decrease* in a number of early warning indicators such as the mean, variance, coefficient of variation ($\sigma/\mu$), index of dispersion ($\sigma^2/\mu$) and autocorrelation. Some countries exhibited a significant *increase* in the skewness and the first differences in the variance. Overall, however, early warning indicators that were found to display notable signal across a number of countries are the mean, variance or combinations thereof. Figures S6–S15 in the electronic supplementary material, show sensitivity analyses for the 10 early warning indicators across different rolling window sizes for detrending and estimation, indicating that the pattern shown in figure 3 is robust to different choices of these hyperparameters.

Table 1 shows the number of significantly rising or falling early warning indicators, the length of the selected time series, the start of the second wave and the respective posterior mean for $R_t$. From theory, we expect all early warning indicators to rise except the coefficient of variation [25], yet we find that most of the indicators show a tendency to fall instead. In the next section, we turn to a simulation study to investigate the possible reasons for this poor performance.

## 3. Early warning signals for COVID-19 in simulation

We conducted simulations to investigate possible reasons that could underlie the poor performance of early warning indicators to anticipate the second COVID-19 wave. In what follows, we first describe the model set-up we use and illustrate how early warning indicators perform under ideal conditions in §3(a). In §3(b), we describe our general simulation set-up, relaxing the separation of timescales to quantify the erosion in performance. We report the simulation results in §3(c).

### (a) Model

We illustrate early warning indicators in the context of a first outbreak that is closely followed by a second one by simulating from a stochastic SEIR model calibrated to COVID-19 using the *pomp* R package [37]. In particular, let $S(t)$, $E(t)$, $I(t)$, $R(t)$ denote the number of individuals in the susceptible, exposed, infectious and recovered compartment at time point $t$, respectively, and let $\Delta N_{S \to E}$, $\Delta N_{E \to I}$ and $\Delta N_{I \to R}$ denote the number of individuals that have transitioned from one compartment to another during the time interval $[t, t + \Delta t]$. The model is updated according to

$$\Delta N_{S \to E} \sim \text{Binomial}\,(S(t), 1 - e^{-\lambda S(t) \Delta t}), \tag{3.1}$$

$$\Delta N_{E \to I} \sim \text{Binomial}\,(E(t), 1 - e^{-\sigma E(t) \Delta t}) \tag{3.2}$$

$$\Delta N_{I \to R} \sim \text{Binomial}\,(I(t), 1 - e^{-\gamma I(t) \Delta t}), \tag{3.3}$$

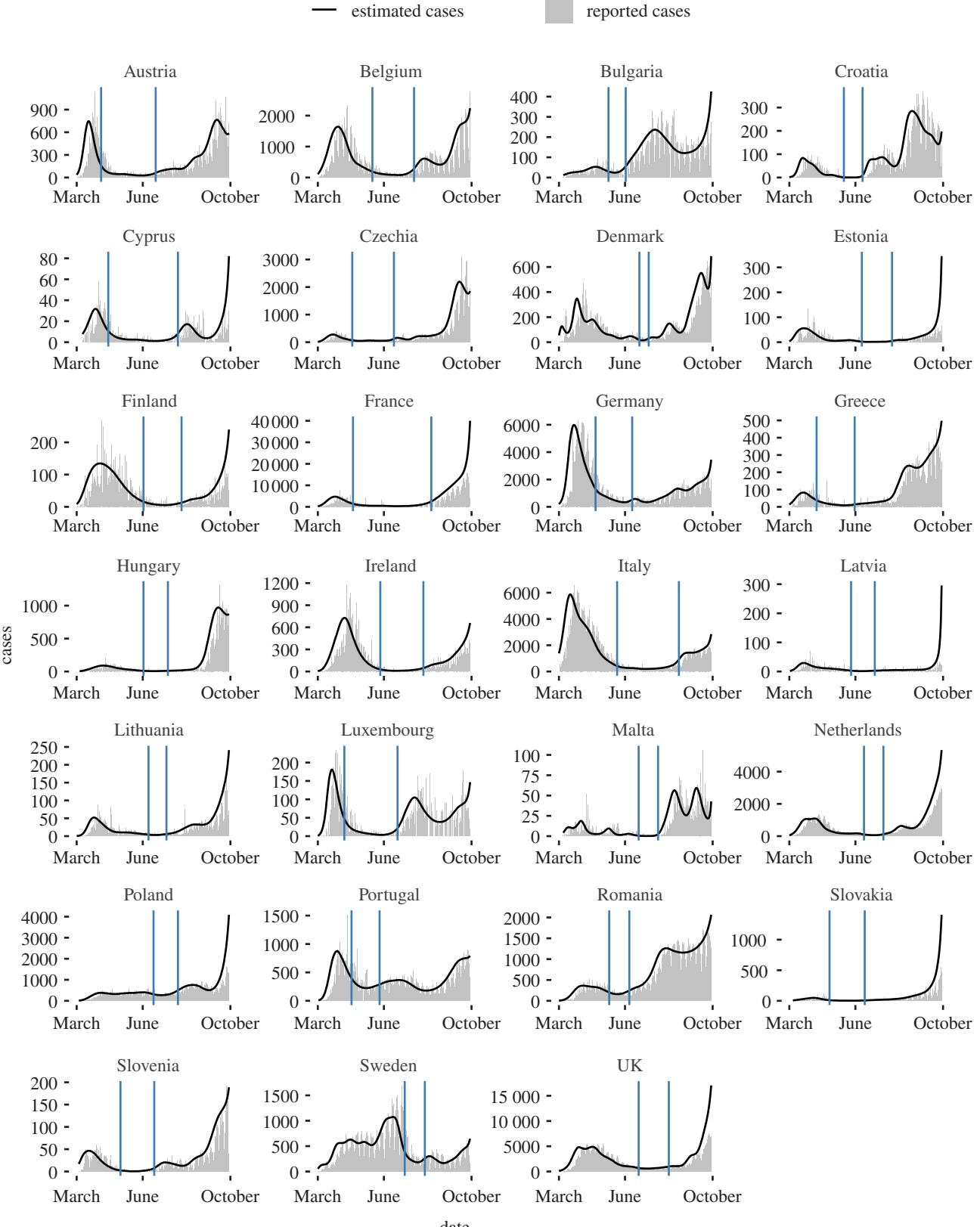

**Figure 2.** Reported cases across European countries. Top: Reported cases (grey) and posterior mean of inferred infected cases (black) for European countries. Vertical blue lines indicate the portion of the time series for which early warning indicators are computed. (Online version in colour.)

where we assume an average incubation and infectious period of $1/\sigma = 5.2$ days [38] and $1/\gamma = 10$ days [39]. The force of infection is given by

$$\lambda = \beta(t)\frac{I(t)}{N} + \eta(t), \tag{3.4}$$

where $\eta(t)$ is the sparking rate, which we assume to be zero until day 50, from which point onward cases are imported with a rate of $\eta = 1/50\,000$. Our goal here is not to produce a simulation model that accurately tracks the COVID-19 outbreak, but instead to investigate critical slowing down in a standardized system that we understand well. To do so, we

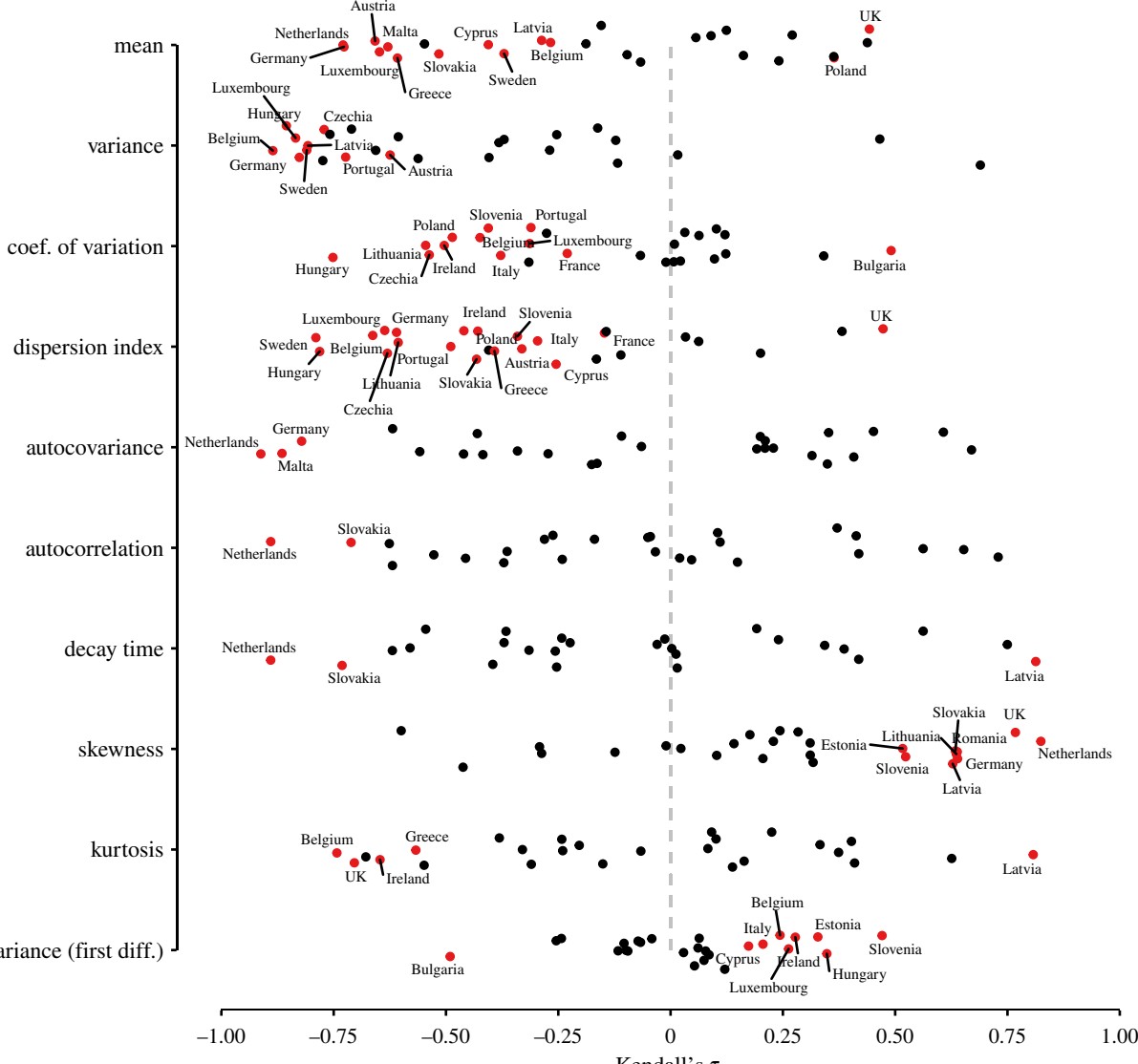

**Figure 3.** Summary of results across countries and indicators. The figure displays Kendall's $\tau$ across European countries for 10 early warning indicators using $\delta_1 = 4$ for detrending and $\delta_2 = 15$ for indicator estimation. Red points indicate countries for which $\tau$ was either significantly smaller or larger than expected under a stationary time series at $\alpha = 0.05$. (Online version in colour.)

wish to force $R_t$ to create a multi-wave epidemic. We achieve this by changing $\beta(t)$ accordingly, compensating for the depletion of the susceptible population by multiplying with $S_0/S(t)$ at time point $t$. This mathematical trick avoids a decrease in $R_t$ over time as the pool of susceptibles gets depleted, and hence allows us to directly manipulate $R_t$. Lastly, we assume that each infected person is reported without delay.

To illustrate the phenomenon of critical slowing down under ideal conditions, we start with 10 000 infected persons out of $N = 1\,000\,000$ and $R_0 = 3$. This results in a first outbreak, which is rapidly brought down through control measures that we model as bringing $R_t$ down to 0.50 within 25 days. We then force $R_t$ to remain at this low value for 200 days, and then allow it to rise linearly to $R_t = 1$, forcing a second wave. This simulation mimics the situation at the start of the pandemic where the first outbreak caught countries by surprise and lockdown was the key mitigation measure that substantially reduced the effective reproductive number. In the illustration, mitigation measures are maintained for a long period of time. However, in reality

mitigation measures were slowly relaxed towards the summer, and with no vaccination in place together with imported infections and increased mixing, the system could not reach a disease-free equilibrium and the reproductive number increased again. This led to a second outbreak in the fall of 2020 in virtually all European countries. Our simple model adequately describes this general pattern as shown in figure 4a. In particular, the left column in figure 4a shows the two waves of transmission and their associated early warning indicators, while the right panels in figure 4a show a similar situation except that no second outbreak occurs ($R_t$ is maintained at low levels). In contrast to the situation with a second wave, the variance and autocorrelation do not rise in this case. This illustration demonstrates that under these conditions a second epidemic wave can be anticipated using non-parametric early warning indicators.

It is known that epidemiological systems can experience a *bifurcation delay*, which describes the transient trajectory of an epidemic as its attracting equilibrium changes. One consequence of bifurcation delays is that the time for a large outbreak to settle to its equilibrium even after crossing $R_t > 1$

**Table 1.** The number of significantly rising or falling early warning indicators, out of a total possible of 10, for European countries together with the length of the selected time series and the respective posterior mean of $R_t$. $\mathcal{D}$ denotes the (country-specific) data, see figure 2.

| country | no. significant ↑ | no. significant ↓ | duration | $\mathbb{E}[R_{min} \mid \mathcal{D}]$ | $\mathbb{E}[R_{max} \mid \mathcal{D}]$ |
|---|---|---|---|---|---|
| Latvia | 3 | 2 | 34 | 0.77 | 1.23 |
| UK | 3 | 1 | 43 | 0.86 | 1.10 |
| Slovenia | 2 | 2 | 48 | 0.63 | 1.48 |
| Estonia | 2 | 0 | 43 | 0.61 | 1.45 |
| Germany | 1 | 6 | 52 | 0.77 | 1.22 |
| Belgium | 1 | 5 | 59 | 0.83 | 1.38 |
| Slovakia | 1 | 5 | 50 | 0.66 | 1.33 |
| Luxembourg | 1 | 4 | 75 | 0.67 | 1.48 |
| The Netherlands | 1 | 4 | 28 | 0.77 | 1.32 |
| Hungary | 1 | 3 | 35 | 0.79 | 1.18 |
| Ireland | 1 | 3 | 61 | 0.72 | 1.28 |
| Cyprus | 1 | 2 | 98 | 0.72 | 1.42 |
| Italy | 1 | 2 | 87 | 0.80 | 1.31 |
| Portugal | 1 | 2 | 40 | 0.82 | 1.07 |
| Bulgaria | 1 | 1 | 25 | 0.84 | 1.31 |
| Romania | 1 | 0 | 29 | 0.87 | 1.14 |
| Austria | 0 | 3 | 77 | 0.63 | 1.25 |
| Czechia | 0 | 3 | 59 | 0.79 | 1.38 |
| Sweden | 0 | 3 | 29 | 0.68 | 1.17 |
| France | 0 | 2 | 110 | 0.77 | 1.27 |
| Greece | 0 | 2 | 54 | 0.81 | 1.19 |
| Lithuania | 0 | 2 | 26 | 0.83 | 1.19 |
| Malta | 0 | 2 | 28 | 0.52 | 2.38 |
| Poland | 0 | 2 | 35 | 0.91 | 1.16 |
| Croatia | 0 | 0 | 27 | 0.38 | 2.85 |
| Denmark | 0 | 0 | 14 | 0.66 | 1.39 |
| Finland | 0 | 0 | 54 | 0.80 | 1.22 |

can be considerable. Dibble *et al.* [18] studied bifurcation delays for disease emergence, and figure 4 indeed shows that it takes a while for the system to show a significant rise in cases even after $R_t > 1$ (see Hungary in figure 2, for a possible example with regards to COVID-19). As can be seen in figure 4, a bifurcation delay also occurs for disease elimination. In particular, for $R_t < 1$ the disease is not endemic and the stable equilibrium consists of a number of new cases that depends on the rate of at which cases are imported. There is, however, a substantial delay between the point at which $R_t < 1$ for the first time and a low number of newly reported cases. This means that early warning indicators computed immediately from the period after $R_t$ first declines to less than 1 would track a transient far from equilibrium and thus would not provide information about the return rate to equilibrium from small perturbations, i.e. the phenomenon of critical slowing down.

To understand the extent to which this bifurcation delay may influence the performance of early warning indicators, we decreased the time interval for which $R_t = 0.50$ from 200 days (figure 4a) to 50 days (figure 4b). We find that both the variance and autocorrelation first *decrease* in the case of both a second outbreak (left panels) and in the case of no second

outbreak (right panels). The variance then rises slightly prior to the second wave, a pattern that does not occur for the autocorrelation, nor for the indicators in case of no second wave. This pattern hints at the fact that the bifurcation delay at elimination will interfere with the detection of critical slowing down if the system is not allowed to settle to its new equilibrium because the magnitude of the transient is commensurate with (or larger than) the magnitude of the fluctuations.

## (b) Simulation set-up

We conducted additional simulations to systematically assess the extent to which these patterns impact the performance of early warning indicators. The forcing of $R_t$ in the previous illustrations depends on five parameters: the value of $R_0$; the value of the lowest point $R_t$ reaches; the time it takes $R_t$ to reach it; the time for which $R_t$ stays at the lowest point and the time it takes $R_t$ to reach criticality again. We again assume that $R_0 = 3$ and that it takes the system 25 days to reach its lowest point of $R_t = 0.50$, but we vary the number of days for which $R_t$ is held constant to be $t_1 \in [25, 50, 100, 200]$ and the time it takes the system to reach $R_t = 1$ to be $t_2 \in [25, 30, \ldots, 95, 200]$. For comparison, we also simulate from a system that stays at $R_t = 0.50$ and

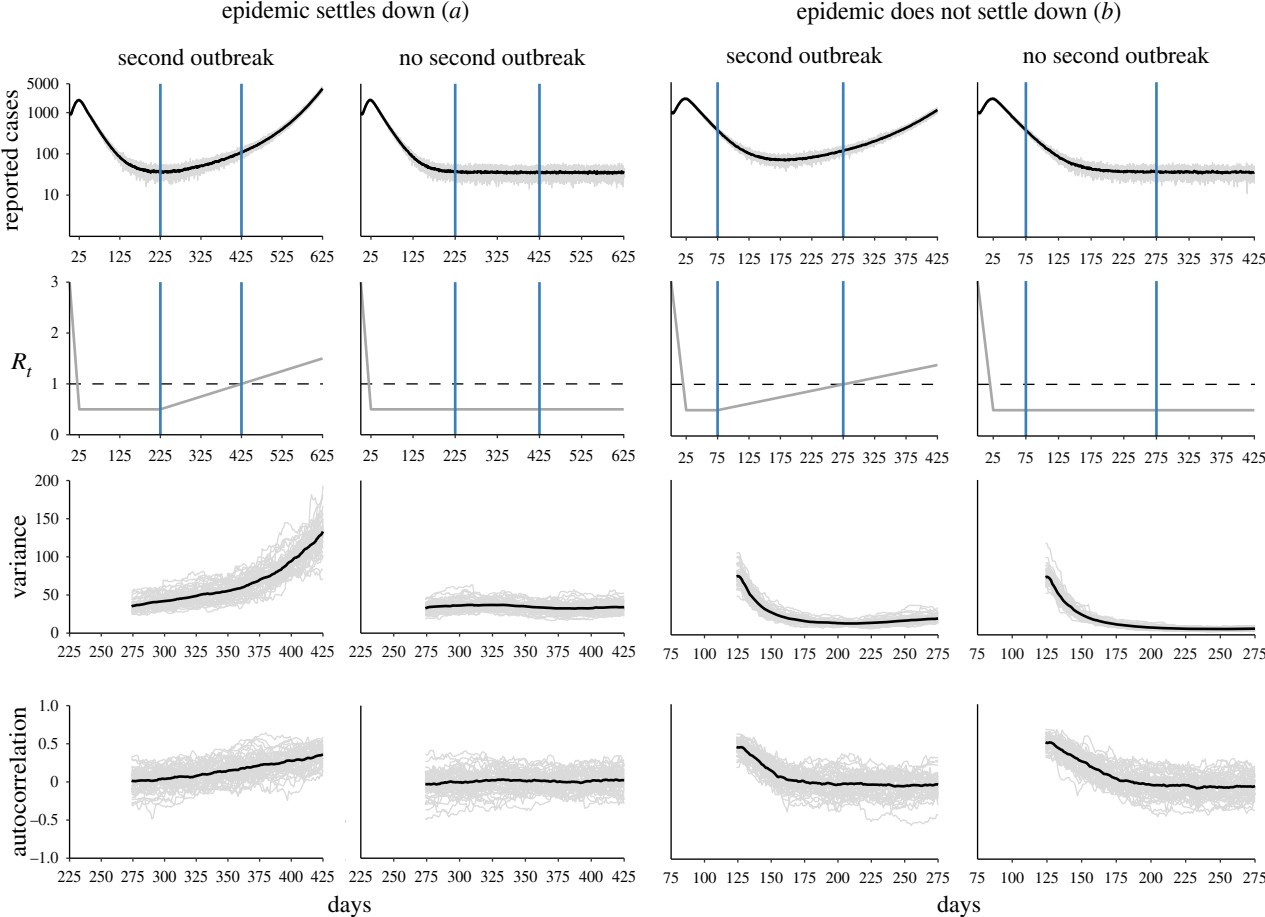

**Figure 4.** Signatures of critical slowing down in a simulated second-wave epidemic. (*a*) Reported cases of a first outbreak followed by a second (top left) or no outbreak (top right) together with the forcing of $R_t$ (below). Vertical blue lines indicate the period on which we compute the early warning indicators autocorrelation and variance, shown in the two bottom panels. The increase in variance and autocorrelation in the left panels is the manifestation of critical slowing down. Shown are 50 simulation runs (grey) together with their mean (black). (*b*) Same, but for the case that the epidemic has not settled down after a first outbreak before a second one is forced. (Online version in colour.)

does not exhibit a second outbreak. We match the length of the time series on which we compute early warning indicators ($t_2$) in case of no outbreak to when an outbreak does occur. As before, backwards rolling windows with a uniform kernel were used for detrending and non-parametric estimation of the mean, variance, coefficient of variation, index of dispersion, skewness, kurtosis, autocovariance, autocorrelation, decay time and first differences in variance. We used rolling windows a 10th the size of the duration for which $R_t$ stays constant; that is, for $t_1 \in [25, 50, 100, 200]$ we used rolling windows of sizes $\delta_1 = [3, 5, 10, 20]$, respectively. For indicator estimation, we used rolling window sizes of $\delta_2 = 25$ (except for the mean), using the R package *spaero* [35]. We simulated 500 trajectories for each setting and calculated the area under the curve (AUC), a measure of classification performance, for all indicators. For each indicator, we calculated its rank correlation with time (Kendall's $\tau$), which indicates whether the early warning indicators rise or fall prior to reaching the critical point. The AUC can then be estimated as the probability that $\tau_{test}$ is larger than $\tau_{null}$ [25,40]. A value of $|AUC - 1/2| = 0$ indicates chance performance, with $AUC < 1/2$ and $AUC > 1/2$ indicating a fall or rise in indicators prior to criticality, respectively. Theory predicts a pre-critical increase of all early warning indicators except the coefficient of variation [19,25]. In addition to AUC, which requires comparing the indicator trend in the case of a second outbreak to the case of no second outbreak, we also use the method proposed by Dakos *et al.* [36] based on ARMA null

models and outlined in §2(*a*) to ascertain whether an indicator rises significantly. This more closely mimics the real-world situation where we do not have access to the counterfactual situation in which no outbreak occurred. We report the true positive rate, that is, the proportion of times we find $p < \alpha$ for each indicator and condition, using $\alpha = 0.05$.

## (c) Simulation results

Figure 5*a* shows that the performance of early warning indicators improves with the time it takes the epidemic to reach a second critical wave. For the case for which the system stays for 200 days at $R_t = 0.50$ (top panel of figure 5*a*), we find that all indicators except the kurtosis and the coefficient of variation performed well, with the mean and the variance performing best. The coefficient of variation, given by the ratio of the standard deviation to the mean, decreases prior to criticality, indicating that the mean rises more quickly than the standard deviation. Most early warning indicators perform worse when $R_t = 0.50$ for 100 days, yet the mean and variance still perform well overall. Interestingly, the slight decrease in performance in the variance implies a stronger decrease of the coefficient of variation. The index of dispersion begins to show a decrease as well when the system is forced more quickly (i.e. $t_2 < 50$).

For a period during which $R_t = 0.50$ of 50 days, the performance of the variance decreases, leading to an

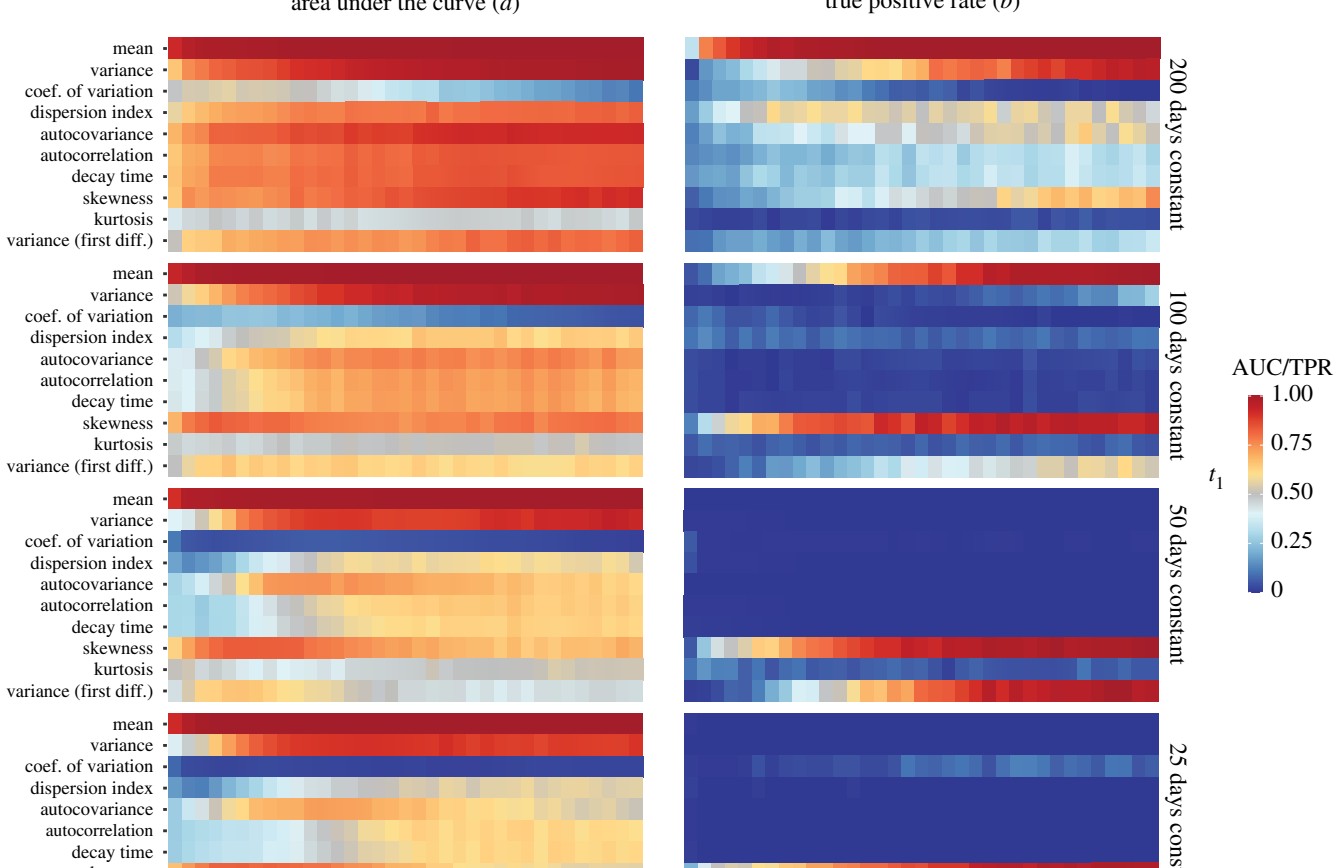

**Figure 5.** Indicator performance across simulation settings. Area under the curve (*a*) and true positive rate (*b*) for 10 early warning indicators as the number of days for which $R_t = 0.50$ and the number of days it takes the system to reach $R_t = 1$ again vary. True positive rate is calculated by using the best-fitting ARMA($p$, $q$) model to create a stationary null distribution and a decision criterion of finding a significant *increase* at $p < 0.05$. (Online version in colour.)

increasingly strong decrease in the coefficient of variation. When forcing is rapid (i.e. $t_2 < 75$), the index of dispersion, autocovariance, autocorrelation, and decay time also begin to show a stronger downward trend (AUC < 1/2) prior to reaching the critical point. These trends are exacerbated when the system stays at $R_t = 0.50$ for only 25 days. One may think that the simulation shows the *reverse* pattern than the empirical analysis, summarized in figure 3, because the mean and variance show a positive AUC (hence they *increase* compared to the null simulation) while the mean and variance show a *decrease* in the empirical analysis. There is no contradiction, however, because the mean and variance do in fact decrease in case of a second wave, it is just that they *decrease less* compared to when there is no second wave, as can be seen in figure 4*b*.

In the data, the median time for countries to go from their minimum $R_t$ value after the first crossing to their maximum $R_t$ value after the crossing was 42 days. Figures S1–S5 in the electronic supplementary materials further show that $R_t$ basically never stays at a low constant value for a sustained period of time, but is forced immediately towards the critical point. Under the most realistic scenario in our simulation study ($t_1 = 25$ and $t_2 < 50$), many indicators perform poorly, yet we still find excellent performance of a rising mean and excellent performance of a falling coefficient of variation and index of dispersion. This does not imply, however, that they will lead

to reliable warnings in practice. While we can quantify discriminatory power using AUC in simulations, in practice early warning indicators have to be *calibrated*. Figure 5*b* shows that testing for an indicator increase at $\alpha = 0.05$ based on a stationary null distribution created by using the best-fitting ARMA($p$, $q$) model to the time series under consideration is poorly calibrated, leading to an extremely poor true positive rate which mirrors the empirical results in §2(*b*). This is because the distribution of Kendall's $\tau$ under the stationary model is centred around zero, while the actually observed Kendall's $\tau$ is negative. As a result, hypothesis tests for an increase in indicator values are expected to suffer from extremely low statistical power in realistic situations. This problem may be exacerbated by a potentially poor fit of the model used to create the null distribution.

## 4. Discussion

Early warning signals based on the phenomenon of critical slowing have been suggested as a way to anticipate transitions in a wide range of dynamical systems, including the (re)emergence of infectious diseases. We analysed whether a suite of indicators could have given early warning of the second COVID-19 wave in European countries. We found that the majority of indicators did not rise reliably, instead showing a

pronounced decrease, a finding inconsistent with previous applications of the theory of critical slowing down. To understand this pattern, we conducted a simulation study in which we varied the time that is available for the system to settle at its new equilibrium after a first outbreak, as well as the speed with which a second wave is forced. We analysed the performance of early warning indicators using the AUC to quantify classification performance and—using the same methodology with which we analysed the empirical data—the true positive rate. We found that classification performance suffered when the system had too little time to settle to its new (quasi-)equilibrium and the second wave is forced quickly (due to changing conditions in the population, such as reduced adherence to control measures), as we saw in the empirical data. Yet we also found that some indicators, such as the mean, continued to perform well (in terms of AUC) in contrast to what we observed in the empirical analysis. Using the same methodology as in the empirical analysis, however, we found a true positive rate of close to zero when testing for an increase in indicators, which is in line with our empirical results.

Our analyses suggest the following conclusions. First, violating a key assumption of early warning indicators based on critical slowing down—namely that the driver ($R_t$) changes slowly compared with the time it takes the system to return to its equilibrium after small external perturbations—dramatically reduces their performance. While this may be expected from theory, our analyses underscore this point and show that early warning indicators cannot be used to anticipate future outbreaks that are quickly forced after an initial wave. Second, as a consequence of the fact that the system is not allowed enough time to settle at its new stable equilibrium after an initial outbreak, the first part of the data used for early warning indicator estimation constitutes a transient. Hence there is a bifurcation delay not only after $R_t$ crosses one from below, as previously observed and studied (e.g. [18]), but also after $R_t$ crosses one from above. If this transient is incorporated into the indicator estimation, then indicators will show a pronounced *decrease* rather than an *increase*. This does not imply, however, that we can use a decrease in indicators as a signal for a future outbreak that quickly follows an initial one, because such a decrease also occurs in case of no outbreak. The poor performance of early warning indicators in our empirical analysis is likely due to a combination of this transient phenomenon and the quick forcing of $R_t$. Third, our simulation study demonstrated that while early warning indicators can yield high discrimination (i.e. a high AUC), in practice they need to be calibrated. We found that the widely used methodology proposed by Dakos *et al.* [36] with decision criterion $p < 0.05$ is poorly calibrated. This leads to poor performance consistent with our empirical results. The key issue is that the sampling distribution created under this methodology is not centred around a negative Kendall's $\tau$ (implying a decreasing trend) but a Kendall's $\tau$ of around zero (implying no trend). Thus the statistical power to reject the null hypothesis of no increase when actually observing a strong decrease in indicators is too low for these tests to be of practical value in realistic situations. Previous research also suggested that indicators can fail in the COVID-19 context [29].

Some limitations of this study should be kept in mind. Our empirical analysis takes the reported number of cases across European countries at face value. While we accounted for reporting delays, we disregarded any issues related to changes in reporting or testing that may affect the estimation of $R_t$. While the flexible method proposed by Abbott *et al.* [32] renders any bias induced by a change of testing transient, any bias may have indeed changed the true value at which $R_t$ crosses one. A more extensive analysis would look at all countries that experienced a second wave. However, we chose to limit ourselves to European countries because of the comparatively good reporting standards and the fact that there is sufficiently large heterogeneity in epidemic trajectories across European countries for the purposes of this study. On a similar note, because the time period between the end of the first and the beginning of the second wave was shorter than the time period it takes the system to settle at its new stable equilibrium after the first wave recedes in virtually all countries, we expect our findings to generalize well to non-European countries. We used an admittedly conservative criterion for date stamping the end of the first wave and the start of the second one to reduce the extent of the transient period we incorporate for indicator estimation. In particular, we chose the day at which $R_t$ reaches its lowest value as starting point for the computation of early warning indicators. If anything, based on our finding that incorporating the transient decreases performance, our choice may be too charitable. We chose the end date for the indicator computation as the day at which $R_t$ reaches its maximum after crossing one so as to increase the number of time points. If anything, this may again have been too generous. At the same time, while the epidemic unfolded quite distinctly in different European countries, $R_t$ never stabilized at a low value and rose quickly after the first outbreak. These are far from the conditions under which to expect a reliable signal in early warning indicators, and our results should not be interpreted as a rejection of their potential in other applications, including other epidemics.

We used backwards rolling windows to avoid the use of data from the 'future', and our results can thus translate to a situation in which indicators are computed in real-time. A critical issue when using non-parametric estimation concerns the choice of the size of the rolling windows [20,36,41]. There is a trade-off between a window size that is too small, where estimation accuracy suffers, and a window size that is too large, where stationarity is (more severely) violated [19]. If a model is available, Dessavre *et al.* [20] find that detrending based on model simulation works well, but this route is unavailable as an epidemic unfolds for which accurate models do not yet exist. Similarly, while Miller *et al.* [21] found that indicator performance was robust to seasonal forcing, the timescale of such seasonal forcing is much longer compared with the movements of $R_t$ that were observed in some European countries, and which hence may have further reduced performance. We have addressed the issue of window size selection by reporting extensive sensitivity analyses. Our finding that indicators poorly anticipate the second COVID-19 wave is robust to different choices.

Critical slowing down is a phenomenon that has primarily been studied in low-dimensional systems. It is prominent in the study of ferromagnetism and the Lenz–Ising model [42], and has been known to proponents of catastrophe theory since at least the 1970s [43]. Wissel [13] suggested critical slowing down as a way to forecast the extinction of a population of rotifers (see also [14,44]). Scheffer *et al.* [8] brought significant attention to the idea of using critical slowing down as an early warning signal which led to a surge of interest across many fields. Yet there is the obvious question of whether we should expect a phenomenon that pertains primarily to low dimensional systems to occur in the high

dimensional real world. Infectious diseases do not spread in homogeneously mixed populations with people being distinct only in terms of whether they are susceptible, exposed, infected or recovered, as our simulation model assumes. Instead, infectious diseases spread between unique individuals on a network that is itself continuously changing. Studying the effect of test sensitivity and frequency on COVID-19 transmission, Larremore *et al.* [45] find essentially no difference between a homogeneous compartment model and an agent-based model that is calibrated to New York City micro-census data. More relevant to our investigation, Brett *et al.* [22] found that early warning indicators based on critical slowing down do indeed rise prior to an outbreak in high-dimensional network and agent-based models.

A related issue with early warning indicators based on critical slowing down concerns the decision criterion. When do we decide that a rise in indicators is 'significant' and constitutes an early warning? In our empirical analysis, we chose a rise in trend to be significant at the $\alpha = 0.05$ level, but this may well require adaption to the specific case at hand. There is a difference between making a statistical inference (e.g. estimating Kendall's $\tau$) and making a decision (e.g. restricting mitigation measures; [46]). The latter requires calibration, which is understudied in the context of early warning indicators based on critical slowing down but essential to use in applications. One can also question the adequacy of the best-fitting ARMA$(p, q)$ model as a null model more broadly. Boettiger & Hastings [46] have shown that statistically comparing two models, one that includes the bifurcation and one that does not, can outperform non-parametric testing using null models (see also [47]). We have used the ARMA null models because they are the most widely used methodology for assessing early warning signals [36] and allow straightforward significance testing for a wide range of indicators. Importantly, some indicators, such as the mean and variance, continue to rise even after $R_t$ crosses one, as predicted by theory [23,48]. Others are expected to peak at the point at which $R_t = 1$, although the exact maximum may not be clear [24]. This means that it is hard to assess whether, say, a rise in the autocorrelation from 0.50 to 0.70 is already problematic, or whether one should wait until it reaches, say, 0.90 (if it ever will). The extent to which indicators such as autocorrelation rise also depends on a number of reporting details such as the frequency of reporting. It is therefore impossible to provide general guidelines for use in applications. Simulation studies that incorporate reporting issues and focus on specific diseases may provide further insight [25,49].

Early warning indicators based on critical slowing down promise to be a quite general and low-cost tool to monitor the emergence and elimination of infectious diseases (e.g. [27,49,50]). It is understudied how well these indicators perform compared to other tools that may be used as early warning signals. In the context of COVID-19, it seems plausible that by making stronger assumptions about the dynamics of the system or using system-external information such as mobility would lead to much better early warning systems. Simply estimating $R_t$ and forecasting whether and when $R_t > 1$ may be a similarly low-cost but potentially more reliable approach. Conceptually, however, it is not so clear that one would like to have an early warning indicator signalling that $R_t$ is about to cross one. This is due to two related reasons. First, because of the bifurcation delay, it may take weeks or months for the actual outbreak to occur. A method that is able to incorporate this bifurcation delay and produce an early warning of an actual exponential increase in cases may therefore be preferable. Ideally, such a method produces a probabilistic assessment of an outbreak, which can then feed into further decision-making. Second, the simple fact that $R_t$ crosses one does not imply that a second wave is incumbent. Instead, it may stay there for a while or fall again, as it did in several European countries during the current pandemic. One cannot impose strong mitigation measures to curb virus spread whenever $R_t > 1$. All this points to a more continuous approach in which multiple, system-external factors are taken into account to assess the risk of future outbreaks. Early warning indicators may be a part of this risk assessment toolbox for (re)emerging diseases when an outbreak is slowly forced—but not, as we have shown, when one outbreak follows closely after another.

Data accessibility. The COVID-19 data used are publicly available from the WHO. The code to reproduce all analyses and figures is available from https://github.com/fdabl/Early-Warning-Covid. The data are provided in electronic supplementary material [51].

Authors' contributions. F.D.: conceptualization, formal analysis, investigation, methodology, software, visualization, writing—original draft, writing—review and editing; H.H. and D.B.: conceptualization, funding acquisition, investigation, supervision, writing—review and editing; J.M.D.: investigation, methodology, supervision, writing—review and editing.

Competing interests. We declare we have no competing interests.

Funding. F.D. and H.H. were supported by ZonMw grant no. 10430022010001. H.H. was also supported by ZonMw grant no. 10430032010011. J.M.D. was supported by NSF grant no. 2027786.

## Endnotes

[1]We analysed countries in the EU, excluding Spain because of a strong weekend reporting effect that presented difficulties for model convergence, as well as the UK.

[2]We use a broad range of indicators to assess the robustness of our conclusions, noting that from a theoretical perspective the variance and autocorrelation are preferred, as they are necessary features of critical slowing down. The variance is especially useful because the divergence should make it highly detectable, whereas incremental changes in the autocorrelation coefficient, which is bounded, will be harder to pick up.

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
