## [Peer Review File · Proceedings of the Royal Society B: Biological Sciences]

Review History

RSPB-2021-1809.R0 (Original submission)

Review form: Reviewer 1

Recommendation

Accept with minor revision (please list in comments)

Scientific importance: Is the manuscript an original and important contribution to its field?

Good

General interest: Is the paper of sufficient general interest?

Good

Quality of the paper: Is the overall quality of the paper suitable?

Excellent

Is the length of the paper justified?

Yes

Should the paper be seen by a specialist statistical reviewer?

No

Do you have any concerns about statistical analyses in this paper? If so, please specify them explicitly in your report.

No

It is a condition of publication that authors make their supporting data, code and materials available - either as supplementary material or hosted in an external repository. Please rate, if applicable, the supporting data on the following criteria.

Is it accessible?

Yes

Is it clear?

Yes

Is it adequate?

Yes

Do you have any ethical concerns with this paper?

No

Comments to the Author

Dablander and colleagues provide an analysis of early warning signals in Covid case data prior to the second wave in European countries. They find that, contrary to expectations of critical slowing down, early warning signals do not generally manifest prior to the second wave. They investigate why this may be the case using a simple disease model. The model demonstrates that if the reproduction number R varies on a time scale similar to that of the internal disease dynamics, then early warning signals cannot be expected prior to disease outbreak. In particular, if R increases towards the threshold before the first wave has had a chance to settle down to an equilibrium state, then indicators can show a decreasing rather than an increasing trend, as observed in the data. They emphasise that early warning signals based on critical slowing down are suitable only if the external forcing of the system varies on a much slower time scale than that of the internal dynamics i.e. time-scale separation is required.

I very much enjoyed reading this paper. It is clearly written and has excellent figures to accompany the narrative. The topic also – early warning signals in the context of Covid – is of significant interest. The authors present the result that early warning metrics based on critical slowing down (as they are currently used) are not appropriate for predicting outbreaks in quick succession of one-another. The work is timely and the scientific approach rigorous, as far as I can see. I therefore recommend this article for publication, though I list some revisions below. Most are minor, however #4 is important to address as it may influence the trend of the indicators, though unlikely the results of the paper.

1. At the beginning of section 2, there is a summary of what the section contains only up to subsection 2.2. A summary of what is done in subsection 2.3+ would be nice. Or, the model and simulation could appear in their own section.

2. Methods: I didn't see any information on preprocessing of the covid data (if any was required?). What was the time series resolution of the data? Were the data evenly spaced, and if not, how was this dealt with?

3. Methods: you mention fitting an ARMA(p,q) model to the data. Do you mean that you fitted several ARMA models for different p and q values (which values?) and then used AIC to determine which p and q values were best? I think this could be made more clear.

4. Methods: Given that you use a backwards rolling window to compute the indicators, it's not clear to me how the values for the indicators at the start of the window of evaluation (blue lines)

are obtained. For example, in Figure 4b, the variance and autocorrelation time series all have values at day 75. Are these values computed by using data from before the window of evaluation i.e. 75-delta_2 up to 75? Or, are they computed using just the single data point at day 75, which would actually explain why the variance is zero on day 75. The standard approach (see e.g. Dakos et al. Methods for Detecting Early Warnings of Critical Transitions in Time Series Illustrated Using Simulated Ecological Data) is to only compute variance and autocorrelation at times where there is enough preceding data to fill the rolling window. So in this example, variance would be computed from day 75+delta_2 up to 275. This may influence the early trend of the indicators, so is important to address.

5. It would be helpful if you indicated t_1 and t_2 on the axes of figure 5. That way when jumping between manuscript and figure it's easier to follow along.

6. Why does the sensitivity analysis for TPR in Figure 5 use a smaller range of t_2 than for the AUC? Can you use the same range for t_2 ?

In the discussion, there is a nice paragraph on how critical slowing down has been mostly studied in low-dimensional systems, and whether we should expect these theories to be applicable to high-dimensional systems. An interesting mathematical note is that high-dimensional systems do tend to exhibit lower-dimensional dynamics as they get close to a bifurcation. The mathematical details are that the dynamics converge onto a lower-dimensional manifold, that is determined by the dominant eigenvectors corresponding to the bifurcation (see e.g. Kuznetsov. Elements of applied bifurcation theory). This is why (in some cases) CSD results from low-dim systems can be extended to high-dim systems.

Review form: Reviewer 2

Recommendation

Accept with minor revision (please list in comments)

Scientific importance: Is the manuscript an original and important contribution to its field?

Good

General interest: Is the paper of sufficient general interest?

Good

Quality of the paper: Is the overall quality of the paper suitable?

Good

Is the length of the paper justified?

Yes

Should the paper be seen by a specialist statistical reviewer?

No

Do you have any concerns about statistical analyses in this paper? If so, please specify them explicitly in your report.

No

It is a condition of publication that authors make their supporting data, code and materials available - either as supplementary material or hosted in an external repository. Please rate, if applicable, the supporting data on the following criteria.

Is it accessible?

Yes

Is it clear?

N/A

Is it adequate?

N/A

Do you have any ethical concerns with this paper?

No

Comments to the Author

Overlapping Time Scales Obscure Early Warning Signals of the Second COVID-19 Wave by Dablander et al

This manuscript presents an application of early warning indicators based on the phenomenon of critical slowing down, to prediction of the second wave of COVID-19 observed in several European countries. The manuscript is well written and I believe the results presented are novel and interesting and will be of interest to the readers of the journal. However, there are a few issues with the manuscript that need to be addressed before it is suitable for publication.

Based on COVID-19 observations prior to the second wave the manuscript demonstrates that a particular approach to applying such early warning indicators can't be used to predict the second wave. A subsequent simulation study shows that the extent to which the epidemic has not reached its new quasi-equilibrium prior to the rise in R_t that produces a second wave, impairs the ability of such early warning indicators to predict a second wave. This is consistent with the underlying theoretical basis for such indicators as it violates the assumption of quasi-equilibrium. In addition, the simulation study shows that the particular approach to assessing significance to trends in indicators used to analyse the COVID-19 data is unable to reliably reveal trends in early warning indicators seen in the simulated examples.

It therefore seems to me that the paper really says that the early warning indicators explored are undermined by transients, but nonetheless might work in real world scenarios like predicting the 2nd wave of COVID-19. And the key unaddressed question is the adequacy of the null model used to identify significant trends in early warning indicators. Whilst I do not think that it is a requirement of publication for the authors to solve this issue, I do think that this needs to be highlighted more clearly. For example, the extent to which the authors think that there is potential to improve on this, or if they believe that this obstacle is insurmountable. Alternatively if the above interpretation of the results is incorrect then some effort needs to be expended to explain why.

In addition below I highlight some further comments, possible typos and suggested re-wordings:

Early on can the authors provide a description of what a suitable (from the point of view of early warning indicators) post 1st wave quasi-equilibrium would look like? Presumably they do not mean disease extinction. Do they just mean sufficiently close to extinction (later it is mentioned the quasi equilibrium results from external infections)?

A number of indicators are assessed but it is not clear that they are all equally valid. Are any to be preferred from a theoretical (or indeed statistical) perspective?

At the start of Section 2 or (e.g. end of 1st para in Section 2.1): it would be good to describe the case data used/reassure the reader that the data provide a reliable basis on which to estimate R_t and the epidemic trajectory. This is alluded to in the discussion, but given the recognised problems with COVID-19 reporting data it would be useful to address this earlier.

The authors state that "Overall, our selection criterion is biased in favor of detecting critical slowing down." However, it is not clear to me why waiting until R_t reaches its maximum (and not just using data when it is increasing at its maximum rate) ensures this, although I accept the practical point of wanting to maximise the amount of data available. Can the authors provide some simulation results to support this assertion?

Just after the equation for S_t : ... for estimation of other estimators e.g. variance etc.

Last para of page 3: An improved description of the use of Kendall's tau would be helpful e.g. is it measuring the rank correlation of indicator value v_t versus t ? Later there is mention of Kendall's tau for 1st differences. Further explanation is also needed to explain why early warning signals calculated from the best fit ARMA(p,q) model generate a suitable null model

Paragraph after Eq 4: The comment about $\beta(t)$ needs some explanation. It seems quite odd to offset susceptible depletion in this way. This leads to a constant R_t and not an increase in R_t and certainly not an R_t profile that that looks like 2nd wave of COVID-19 shown in Fig 1 as implied by the current wording (We achieve this ...). Is this adjustment really needed and if so why?

2nd para page 8: Add comment in brackets: ... while the right panels in Figure 4a show a similar situation except that no second outbreak occurs (R_t is maintained at the low level)

2nd para page 8: suggested rewording: ... after R_t first declines to less than 1, *would track* a transient far from equilibrium and *thus would not* provide information about the ...

Section 2.4 Simulation setup: Given the previous paragraph shows simulation results would this section be better described as, Sensitivity Analysis?

Para 1, page 10: "For indicator estimation, we used rolling window sizes of 50 ...". Doesn't this wording (also as used above) imply the mean is not an indicator?

End of para 1, page 10: Please add some more detail to help the reader identify what method is referred to by "and outlined in Section 2.1 to ascertain ..."

Section 2.5 (page 10) Fig 5a, top panel: When discussing the coefficient of variation say that it's trend is opposite to the other indicators and hence is shown to perform well

Section 3 - Discussion, Rephrase: We analyzed the performance of early warning indicators using the area under the curve to quantify classification performance and using the same methodology with which we analyzed the empirical data, the true positive rate ...

Decision letter (RSPB-2021-1809.R0)

02-Nov-2021

Dear Dr Dablander:

Your manuscript has now been peer reviewed and the reviews have been assessed by an Associate Editor. The reviewers' comments (not including confidential comments to the Editor) and the comments from the Associate Editor are included at the end of this email for your reference. As you will see, the reviewers and the Editors have raised some concerns with your manuscript and we would like to invite you to revise your manuscript to address them.

Research ethics:

Use of animals and field studies:

It is a condition of publication that you make available the data and research materials supporting the results in the article. Please see our Data Sharing Policies (<https://royalsociety.org/journals/authors/author-guidelines/#data>). Datasets should be deposited in an appropriate publicly available repository and details of the associated accession number, link or DOI to the datasets must be included in the Data Accessibility section of the article (<https://royalsociety.org/journals/ethics-policies/data-sharing-mining/>). Reference(s) to datasets should also be included in the reference list of the article with DOIs (where available).

Please submit a copy of your revised paper within three weeks. If we do not hear from you within this time your manuscript will be rejected. If you are unable to meet this deadline please let us know as soon as possible, as we may be able to grant a short extension.

Best wishes,
Dr Locke Rowe
mailto:proceedingsb@royalsociety.org

Associate Editor

Comments to Author:

The reviewers find the paper to be very interesting and well written (and I agree). However, they raise some good points about the overall conclusion of the study (reviewer 2) as well as details of the methods that should be clarified. I think that these points would be fully addressed in the next version of the manuscript.

Reviewer(s)' Comments to Author:

Referee: 1

Comments to the Author(s)

Dablander and colleagues provide an analysis of early warning signals in Covid case data prior to the second wave in European countries. They find that, contrary to expectations of critical slowing down, early warning signals do not generally manifest prior to the second wave. They investigate why this may be the case using a simple disease model. The model demonstrates that if the reproduction number R varies on a time scale similar to that of the internal disease dynamics, then early warning signals cannot be expected prior to disease outbreak. In particular, if R increases towards the threshold before the first wave has had a chance to settle down to an equilibrium state, then indicators can show a decreasing rather than an increasing trend, as observed in the data. They emphasise that early warning signals based on critical slowing down are suitable only if the external forcing of the system varies on a much slower time scale than that of the internal dynamics i.e. time-scale separation is required.

I very much enjoyed reading this paper. It is clearly written and has excellent figures to accompany the narrative. The topic also – early warning signals in the context of Covid – is of significant interest. The authors present the result that early warning metrics based on critical slowing down (as they are currently used) are not appropriate for predicting outbreaks in quick succession of one-another. The work is timely and the scientific approach rigorous, as far as I can see. I therefore recommend this article for publication, though I list some revisions below. Most

are minor, however #4 is important to address as it may influence the trend of the indicators, though unlikely the results of the paper.

1. At the beginning of section 2, there is a summary of what the section contains only up to subsection 2.2. A summary of what is done in subsection 2.3+ would be nice. Or, the model and simulation could appear in their own section.
2. Methods: I didn't see any information on preprocessing of the covid data (if any was required?). What was the time series resolution of the data? Were the data evenly spaced, and if not, how was this dealt with?
3. Methods: you mention fitting an ARMA(p,q) model to the data. Do you mean that you fitted several ARMA models for different p and q values (which values?) and then used AIC to determine which p and q values were best? I think this could be made more clear.
4. Methods: Given that you use a backwards rolling window to compute the indicators, it's not clear to me how the values for the indicators at the start of the window of evaluation (blue lines) are obtained. For example, in Figure 4b, the variance and autocorrelation time series all have values at day 75. Are these values computed by using data from before the window of evaluation i.e. 75-delta_2 up to 75? Or, are they computed using just the single data point at day 75, which would actually explain why the variance is zero on day 75. The standard approach (see e.g. Dakos et al. Methods for Detecting Early Warnings of Critical Transitions in Time Series Illustrated Using Simulated Ecological Data) is to only compute variance and autocorrelation at times where there is enough preceding data to fill the rolling window. So in this example, variance would be computed from day 75+delta_2 up to 275. This may influence the early trend of the indicators, so is important to address.
5. It would be helpful if you indicated t_1 and t_2 on the axes of figure 5. That way when jumping between manuscript and figure it's easier to follow along.
6. Why does the sensitivity analysis for TPR in Figure 5 use a smaller range of t_2 than for the AUC? Can you use the same range for t_2?

In the discussion, there is a nice paragraph on how critical slowing down has been mostly studied in low-dimensional systems, and whether we should expect these theories to be applicable to high-dimensional systems. An interesting mathematical note is that high-dimensional systems do tend to exhibit lower-dimensional dynamics as they get close to a bifurcation. The mathematical details are that the dynamics converge onto a lower-dimensional manifold, that is determined by the dominant eigenvectors corresponding to the bifurcation (see e.g. Kuznetsov. Elements of applied bifurcation theory). This is why (in some cases) CSD results from low-dim systems can be extended to high-dim systems.

Referee: 2

Comments to the Author(s)

Overlapping Time Scales Obscure Early Warning Signals of the Second COVID-19 Wave by Dablander et al

This manuscript presents an application of early warning indicators based on the phenomenon of critical slowing down, to prediction of the second wave of COVID-19 observed in several European countries. The manuscript is well written and I believe the results presented are novel and interesting and will be of interest to the readers of the journal. However, there are a few issues with the manuscript that need to be addressed before it is suitable for publication.

Based on COVID-19 observations prior to the second wave the manuscript demonstrates that a particular approach to applying such early warning indicators can't be used to predict the second wave. A subsequent simulation study shows that the extent to which the epidemic has not

reached its new quasi-equilibrium prior to the rise in R_t that produces a second wave, impairs the ability of such early warning indicators to predict a second wave. This is consistent with the underlying theoretical basis for such indicators as it violates the assumption of quasi-equilibrium. In addition, the simulation study shows that the particular approach to assessing significance to trends in indicators used to analyse the COVID-19 data is unable to reliably reveal trends in early warning indicators seen in the simulated examples.

It therefore seems to me that the paper really says that the early warning indicators explored are undermined by transients, but nonetheless might work in real world scenarios like predicting the 2nd wave of COVID-19. And the key unaddressed question is the adequacy of the null model used to identify significant trends in early warning indicators. Whilst I do not think that it is a requirement of publication for the authors to solve this issue, I do think that this needs to be highlighted more clearly. For example, the extent to which the authors think that there is potential to improve on this, or if they believe that this obstacle is insurmountable. Alternatively if the above interpretation of the results is incorrect then some effort needs to be expended to explain why.

In addition below I highlight some further comments, possible typos and suggested re-wordings:

Early on can the authors provide a description of what a suitable (from the point of view of early warning indicators) post 1st wave quasi-equilibrium would look like? Presumably they do not mean disease extinction. Do they just mean sufficiently close to extinction (later it is mentioned the quasi equilibrium results from external infections)?

A number of indicators are assessed but it is not clear that they are all equally valid. Are any to be preferred from a theoretical (or indeed statistical) perspective?

At the start of Section 2 or (e.g. end of 1st para in Section 2.1): it would be good to described the case data used/reassure the reader that the data provide a reliable basis on which to estimate R_t and the epidemic trajectory. This is alluded to in the discussion, but given the recognised problems with COVID-19 reporting data it would be useful to address thus earlier.

The authors state that "Overall, our selection criterion is biased in favor of detecting critical slowing down." However, it is not clear to me why waiting until R_t reaches its maximum (and not just using data when it is increasing at its maximum rate) ensures this, although I accept the practical point of wanting to maximise the amount of data available. Can the authors provide some simulation results to support this assertion?

Just after the equation for S_t : ... for estimation of other estimators e.g. variance etc.

Last para of page 3: An improved description of the use of Kendall's tau would be helpful e.g. is it measuring the rank correlation of indicator value v_t versus t ? Later there is mention of Kendall's tau for 1st differences. Further explanation is also needed to explain why early warning signals calculated from the best fit ARMA(p,q) model generate a suitable null model

Paragraph after Eq 4: The comment about $\beta(t)$ needs some explanation. It seems quite odd to offset susceptible depletion in this way. This leads to a constant R_t and not an increase in R_t and certainly not an R_t profile that that looks like 2nd wave of COVID-19 shown in Fig 1 as implied by the current wording (We achieve this ...). Is this adjustment really needed and if so why?

2nd para page 8: Add comment in brackets: ... while the right panels in Figure 4a show a similar situation except that no second outbreak occurs (R_t is maintained at the low level)

2nd para page 8: suggested rewording: ... after R_t first declines to less than 1, *would track* a transient far from equilibrium and *thus would not* provide information about the ...

Section 2.4 Simulation setup: Given the previous paragraph shows simulation results would this section be better described as, Sensitivity Analysis?

Para 1, page 10: "For indicator estimation, we used rolling window sizes of 50 ...". Doesn't this wording (also as used above) imply the mean is not an indicator?

End of para 1, page 10: Please add some more detail to help the reader identify what method is referred to by "and outlined in Section 2.1 to ascertain ..."

Section 2.5 (page 10) Fig 5a, top panel: When discussing the coefficient of variation say that it's trend is opposite to the other indicators and hence is shown to perform well

Section 3 - Discussion, Rephrase: We analyzed the performance of early warning indicators using the area under the curve to quantify classification performance and using the same methodology with which we analyzed the empirical data, the true positive rate ...

Author's Response to Decision Letter for (RSPB-2021-1809.R0)

See Appendix A.

Decision letter (RSPB-2021-1809.R1)

13-Jan-2022

Dear Dr Dablander

I am pleased to inform you that your manuscript entitled "Overlapping Time Scales Obscure Early Warning Signals of the Second COVID-19 Wave" has been accepted for publication in Proceedings B.

Data Accessibility section

Open Access

You are invited to opt for Open Access, making your freely available to all as soon as it is ready for publication under a CC BY licence. Our article processing charge for Open Access is £1700. Corresponding authors from member institutions

Paper charges

Sincerely,
Dr Locke Rowe
Editor, Proceedings B
mailto: proceedingsb@royalsociety.org

Appendix A

Fabian Dablander
University of Amsterdam
Psychological Methods
Nieuwe Achtergracht 129-B
1018 WT, Amsterdam
The Netherlands

December 7, 2021

Dear Dr. Rowe,

We are delighted to hear that the associate editor and reviewers valued our work. We want to thank them for their time and their constructive comments. We have now revised our manuscript in accordance with their suggestions.

The biggest change we made was in response to Reviewer I, who noted that using the data points within the first rolling window may lead to bias in the first few indicator estimates. We therefore repeated all our analyses using the Dakos et al. (2012) approach, which leaves out a specified number of initial data points and which, as the reviewer points out, is the more widely used approach.

We found virtually the same results with only a slight difference. In particular, the coefficient of variation (σ/μ) and the index of dispersion (σ^2/μ) are not increasing anymore, but decreasing. This is indeed because of a bias in the early estimates: the variance is smaller than the mean at the beginning (because it is computed on fewer data points, as visible in e.g. our previous Figure 1) and that leads to a decrease in the coefficient of variation and the index of dispersion since $\mu > \sigma$. The increase in these two indicators was therefore an artefact of our previous approach. Using the more widely used approach by Dakos et al. (2012) – as the reviewer suggested – yields a more consistent decrease in the indicators.

Because the Dakos et al. (2012) approach does not use the first batch of data that has a length smaller than the rolling window, we use a slightly smaller rolling window for indicator estimation in the empirical analysis. Instead of $\delta_2 = 25$, which would gloss over the first 25 data points, we use $\delta_2 = 15$, which glosses over only the first 15. We have updated Table 1 and Figure 3 accordingly. The results are robust to the exact choice, and we report a sensitivity analysis with estimates for $\delta_2 = [5, 6, \dots, 30]$ in the appendix. This differs from the

previous analysis, which reported values in the range $[5, 10, \dots, 50]$. We only show values up to 30 because of data constraints (for many countries, there is no estimate for $\delta_2 > 30$ because the Dakos et al. approach does not use data points smaller than the rolling window) and because larger rolling windows actually make less sense theoretically – the estimation is using too long a period which strongly violates stationarity.

We selectively mention these changes here because they are the biggest ones. That said, we have addressed all comments made by the reviewer and incorporated their helpful suggestions into the manuscript. We detail our responses to each comment and suggestion below. We have also carefully read the manuscript again and made minor textual improvements where necessary. All textual changes in the manuscript are marked in blue.

We want to thank the associate editor and both reviewers again for their time and their constructive comments and hope that our revised version is suitable for publication in *Proceedings of the Royal Society B: Biological Sciences*.

Kind regards,

Fabian Dablander, Hans Heesterbeek,
Denny Borsboom, John Drake

Reviewer 1

Comment 0

“Dablander and colleagues provide an analysis of early warning signals in Covid case data prior to the second wave in European countries. They find that, contrary to expectations of critical slowing down, early warning signals do not generally manifest prior to the second wave. They investigate why this may be the case using a simple disease model. The model demonstrates that if the reproduction number R varies on a time scale similar to that of the internal disease dynamics, then early warning signals cannot be expected prior to disease outbreak. In particular, if R increases towards the threshold before the first wave has had a chance to settle down to an equilibrium state, then indicators can show a decreasing rather than an increasing trend, as observed in the data. They emphasise that early warning signals based on critical slowing down are suitable only if the external forcing of the system varies on a much slower time scale than that of the internal dynamics i.e. time-scale separation is required.

I very much enjoyed reading this paper. It is clearly written and has excellent figures to accompany the narrative. The topic also—early warning signals in the context of Covid—is of significant interest. The authors present the result that early warning metrics based on critical slowing down (as they are currently used) are not appropriate for predicting outbreaks in quick succession of one-another. The work is timely and the scientific approach rigorous, as far as I can see. I therefore recommend this article for publication, though I list some revisions below. Most are minor, however #4 is important to address as it may influence the trend of the indicators, though unlikely the results of the paper.”

We are happy that the reviewer enjoyed reading our paper and finds it suitable for publication. We address the comments in detail below, and wish to thank the reviewer for the constructive comments.

Comment 1

“At the beginning of section 2, there is a summary of what the section contains only up to subsection 2.2. A summary of what is done in subsection 2.3+ would be nice. Or, the model and simulation could appear in their own section.”

We thank the reviewer – this is a good catch. We have now created a separate Section for the simulation study and describe what is discussed there. In particular, we now write:

“We conducted simulations to investigate possible reasons that could underlie the poor performance of early warning indicators to anticipate the second COVID-19 wave. In what follows, we first describe the model set-up we use and illustrate how early warning indicators perform under ideal conditions in Section 3.1. In Section 3.2, we describe our general simulation set-up, relaxing the separation of time scales to quantify the erosion in performance. We report the simulation results in Section 3.3.”

Comment 2

“Methods: I didn’t see any information on preprocessing of the covid data (if any was required?). What was the time series resolution of the data? Were the data evenly spaced, and if not, how was this dealt with?”

We have expanded on this in the methods section, adding:

“We applied this method to a broad range of European countries using daily case report data from the WHO, spanning the period from March to October 2020. We used the R package *covidregionaldata* to load the raw data (Palmer et al., 2021); no further preprocessing was necessary.

Comment 3

“Methods: you mention fitting an ARMA(p,q) model to the data. Do you mean that you fitted several ARMA models for different p and q values (which values?) and then used AIC to determine which p and q values were best? I think this could be made more clear.”

We have clarified this and now write:

“To create a sampling distribution under the null hypothesis of no increase in the early warning indicators that respects the temporal ordering of the data, we fitted a series of ARMA(p, q) models with $(p, q) \in [1, 2, \dots, 5]$ to the country-specific data. We selected the best-fitting model using AIC and subsequently generated 500 surrogate time series from it, computed the early warning indicators as outlined above, and estimated the rank correlation of the indicator values s_t with time t , known as Kendall’s τ .”

Comment 4

“Methods: Given that you use a backwards rolling window to compute the indicators, it’s not clear to me how the values for the indicators at the start of the window of evaluation (blue lines) are obtained. For example, in Figure 4b, the variance and autocorrelation time series all have values at day 75. Are these values computed by using data from before the window of evaluation i.e. $75-\delta_2$ up to 75? Or, are they computed using just the single data point at day 75, which would actually explain why the variance is zero on day 75. The standard approach (see e.g. Dakos et al. Methods for Detecting Early Warnings of Critical Transitions in Time Series Illustrated Using Simulated Ecological Data) is to only compute variance and autocorrelation at times where there is enough preceding data to fill the rolling window. So in this example, variance would be computed from day $75+\delta_2$ up to 275. This may influence the early trend of the indicators, so is important to address.”

Yes, this is the difference between the Dakos et al. approach and the approach developed by John Drake’s group available in the R package *spaeero*. As discussed above, we switched to the Dakos et al. approach. It yields very similar results and conclusions as our initial approach, but without the above-mentioned artefact of a decrease in the coefficient of variation and index of dispersion. We thank the reviewer for pointing us to this.

Comment 5

“It would be helpful if you indicated t_1 and t_2 on the axes of figure 5. That way when jumping between manuscript and figure it’s easier to follow along.”

We have done so now.

Comment 6

“Why does the sensitivity analysis for TPR in Figure 5 use a smaller range of t_2 than for the AUC? Can you use the same range for t_2 ?”

We initially used a smaller range because of the substantially increased computational burden of estimating the ARMA models that is required for calculating the TPR. However, we have now run the simulation study again with the same range t_2 and updated Figure 5 accordingly.

Comment 7

“In the discussion, there is a nice paragraph on how critical slowing down has been mostly studied in low-dimensional systems, and whether we should expect these theories to be applicable to high-dimensional systems. An interesting mathematical note is that high-dimensional systems do tend to exhibit lower-dimensional dynamics as they get close to a bifurcation. The mathematical details are that the dynamics converge onto a lower-dimensional manifold, that is determined by the dominant eigenvectors corresponding to the bifurcation (see e.g. Kuznetsov. Elements of applied bifurcation theory). This is why (in some cases) CSD results from low-dim systems can be extended to high-dim systems.”

The reviewer responds to a sentence in our discussion where we raise the question about results from low-dimensional systems being relevant for (typically) high-dimensional real-world systems. The reviewer highlights the interesting issue that insights into critical slowing down obtained for low-dimensional systems may be relevant for high-dimensional systems as well, as the latter can exhibit lower-dimensional dynamic behaviour when nearing a bifurcation point or an equilibrium. This is correct and the reviewer probably refers to center manifolds. These can indeed be found in the book by Yuri Kuznetsov. One can argue, however, that this set-up differs from the problem we study. The center manifolds relate to asymptotic (attracting) behaviour in dynamical systems, where in our case we are dealing with transient behaviour, particularly after the first wave. Although the problem of relating low-dimensional results to high-dimensional dynamics is an interesting one, we feel further discussion of that in the context of our applied analysis is beyond the scope and calls for a more generic mathematical study.

Reviewer 2

Comment 0

“This manuscript presents an application of early warning indicators based on the phenomenon of critical slowing down, to prediction of the second wave of COVID-19 observed in several European countries. The manuscript is well written and I believe the results presented are novel and interesting and will be of interest to the readers of the journal. However, there are a few issues with the manuscript that need to be addressed before it is suitable for publication.”

We thank the reviewer for the kind words about our work and for the helpful comments, which we address below.

Comment 1

“Based on COVID-19 observations prior to the second wave the manuscript demonstrates that a particular approach to applying such early warning indicators can’t be used to predict the second wave. A subsequent simulation study shows that the extent to which the epidemic has not reached its new quasi-equilibrium prior to the rise in R_t that produces a second wave, impairs the ability of such early warning indicators to predict a second wave. This is consistent with the underlying theoretical basis for such indicators as it violates the assumption of quasi-equilibrium. In addition, the simulation study shows that the particular approach to assessing significance to trends in indicators used to analyse the COVID-19 data is unable to reliably reveal trends in early warning indicators seen in the simulated examples.

It therefore seems to me that the paper really says that the early warning indicators explored are undermined by transients, but nonetheless might work in real world scenarios like predicting the 2nd wave of COVID-19. And the key unaddressed question is the adequacy of the null model used to identify significant trends in early warning indicators. Whilst I do not think that it is a requirement of publication for the authors to solve this issue, I do think that this needs to be highlighted more clearly. For example, the extent to which the authors think that there is potential to improve on this, or if they believe that this obstacle is insurmountable. Alternatively if the above interpretation of the results is incorrect then some effort needs to be expended to explain why. ”

We thank the reviewer for the thoughtful comments, but we are a bit confused by the sentence that it “might nonetheless work in real world scenarios like predicting the 2nd wave of COVID-19”. We show empirically that it does not work for this case, and also illustrate why. The point about the adequacy of the null models is well taken, however. We have expanded on the null model further in the discussion, writing:

“One can also question the adequacy of the best-fitting ARMA(p, q) model as a null model more broadly. Boettiger Hastings (2012) have shown that statistically comparing two models, one that includes the bifurcation and one that does not, can outperform nonparametric testing using null models (see also Bury et al., 2021). We have used the ARMA null models because they are the most widely used methodology for assessing early warning signals (Dakos et al., 2012) and allow straightforward significance testing for a wide range of indicators.”

Comment 2

“In addition below I highlight some further comments, possible typos and suggested re-wordings.

Early on can the authors provide a description of what a suitable (from the point of view of early warning indicators) post 1st wave quasi-equilibrium would look like? Presumably they do not mean disease extinction. Do they just mean sufficiently close to extinction (later it is mentioned the quasi equilibrium results from external infections)?”

The reviewer is correct – we do not mean extinction. Instead, we mean that the system is subcritical ($R_t < 1$) due to imposed measures, but the virus does not go extinct because of the importation of cases. We have added the following sentence to the introduction to make this clear early on: “The quasi-equilibrium corresponds to the dynamics of the epidemic being subcritical ($R_t < 1$) but not dying out due to the importation of cases, instead reaching a quasi-stationary state.”

Comment 3

“A number of indicators are assessed but it is not clear that they are all equally valid. Are any to be preferred from a theoretical (or indeed statistical) perspective?”

We use this broad range of indicators to assess the robustness of our conclusions, and because they have been shown to increase prior to the epidemic transition (except for the coefficient of variation; see Brett et al., 2018). That said, the variance and autocorrelation are the necessary features of critical slowing down, easy to compute, and well understood, and hence are preferred. Moreover, the variance is especially useful because the divergence should make it highly detectable (whereas incremental changes in autocorrelation coefficient, which is bounded, will be hard to pick up). We have added the following footnote to the section where we first mention the indicators:

“We use a broad range of indicators to assess the robustness of our conclusions, noting that from a theoretical perspective the variance and autocorrelation are preferred, as they are necessary features of critical slowing down. The variance is especially useful because the divergence should make it highly detectable, whereas incremental changes in the autocorrelation coefficient, which is bounded, will be harder to pick up.”

Comment 4

“At the start of Section 2 or (e.g. end of 1st para in Section 2.1): it would be good to described the case data used/reassure the reader that the data provide a reliable basis on which to estimate R_t and the epidemic trajectory. This is alluded to in the discussion, but given the recognised problems with COVID-19 reporting data it would be useful to address thus earlier.”

In response to a similar remark by Review 1, we have added the sentence below. We note that the WHO data are the basis for countless studies on the topic, and is the best data source there is.

“We applied this method to a broad range of European countries using daily case report data from the WHO, spanning the period from March to October 2020. We used the R package *covidregionaldata* to load the raw data (Palmer et al., 2021); no further preprocessing was necessary.”

Comment 5

“The authors state that ”Overall, our selection criterion is biased in favor of detecting critical slowing down.” However, it is not clear to me why waiting until R_t reaches its maximum (and not just using data when it is increasing at its maximum rate) ensures this, although I accept the practical point of wanting to maximise the amount of data available. Can the authors provide some simulation results to support this assertion?”

We thank the reviewer for pointing the finger at this. Our original intuition for this statement was that it makes sense to maximize the length of the data, but we realize that we do not have strong evidence that this maximizes the possibility to detect critical slowing down. However, there are many other, less objective choices to be made here and it is unclear a priori which choices would improve performance. We therefore deleted this sentence. We also changed a previous sentence, which now states:

“Similarly, we chose to end the interval with the maximum of R_t after it crosses the threshold as a principled approach that could be systematically applied to all data yielding the longest time series.”

Comment 6

“Just after the equation for S_t : ... for estimation of other estimators e.g. variance etc.”

We do not understand this remark. There is no such sentence after the equation for s_t . The equation is also specific for the variance, so we would like to refrain from mentioning other indicators at this point. We refer to Brett et al. (2018) for the definition of all indicators we use.

Comment 7

“Last para of page 3: An improved description of the use of Kendall’s tau would be helpful e.g. is it measuring the rank correlation of indicator value v_t versus t ? Later there is mention of Kendall’s tau for 1st differences. Further explanation is also needed to explain why early warning signals calculated from the best fit ARMA(p,q) model generate a suitable null model.”

In response to a similar remark by Reviewer 1, we have expanded the description of the ARMA(p, q) null model. We have now also updated the description of Kendall’s tau to reflect the reviewer’s suggestion, writing:

“We selected the best fitting model using AIC and subsequently generated 500 surrogate time-series from it, computed the early warning indicators as outlined above, and estimated the rank correlation with the indicator values s_t with time t , known as Kendall’s τ .”

We have also added the following sentence: “This is the most widely used approach when estimating and testing early warning indicators using rolling windows (Dakos et al., 2012)”, which reflects that we adhere to the approach that has become the standard in the nonparametric estimation of early warning indicators. We note that we have added further reflections on the adequacy of the null model in the discussion as a response to the reviewer’s previous point.

Comment 8

“Paragraph after Eq 4: The comment about $\beta(t)$ needs some explanation. It seems quite odd to offset susceptible depletion in this way. This leads to a constant R_t and not an increase in R_t and certainly not an R_t profile that that looks like 2nd wave of COVID-19 shown in Fig 1 as implied by the current wording (We achieve this ...). Is this adjustment really needed and if so why? ”

We think we may have been unclear in our communication. Our approach does not lead to a “constant R_t ” (see also Figure 4, which illustrates this). In contrast, the approach has the desired effect of a rising value of R_t . There are two mechanisms at play here. The first is that β changes because measures change (or the adherence to measures). We mimic that by forcing β to rise linearly in the model. The second mechanism is that as a result of new infections, R_t decreases slowly as $S(t)$ decreases. Adjusting by $S(0) / S(t)$ is a mathematical trick that makes it so that we can directly manipulate only the β part of R_t relating to the control measures. Because we want to “hard-code” R_t , we do not want the dependency of $R(t)$ on a depleting pool of susceptibles, which would cause R_t to be lower than what we program. Thus, we adjust by $S(0) / S(t)$.

We note that this is a minor adjustment in the period after the first wave because the control measures imposed early on in the pandemic have severely limited the potential number of cases and thus almost the entire population is still susceptible after the initial wave.

We have added the following clarifying remark after the sentence starting with “We achieve [...]”: “This mathematical trick avoids a decrease in R_t over time as the pool of susceptibles gets depleted, and hence allows us to directly manipulate R_t .”

Comment 9

“2nd para page 8: Add comment in brackets: ... while the right panels in Figure 4a show a similar situation except that no second outbreak occurs (R_t is maintained at the low level)”

We have added this comment, thank you!

Comment 10

“2nd parapage 8: suggested rewording: ... after R_t first declines to less than 1, *would track* a transient far from equilibrium and *thus would not* provide information about the ...”

We have implemented this suggested change.

Comment 11

“Section 2.4 Simulation setup: Given the previous paragraph shows simulation results would this section be better described as, Sensitivity Analysis?”

Section 2.3 describes the model and illustrates on a few selected parameter combinations how the lack of time-scale separation influences the performance of early warning signals. Only in Section 2.4 do we perform an extensive simulation study exploring a much larger parameter space. We thus prefer to keep the naming as is. See also our response to comment 1 by reviewer 1.

Comment 12

“Para 1, page 10: ”For indicator estimation, we used rolling window sizes of 50 ...”. Doesn’t this wording (also as used above) imply the mean is not an indicator?”

For the mean, we indeed use a smaller rolling window size. To clarify, we have added the following qualifier in a bracket: “(except for the mean)”. In a sentence before, we also add th qualifier: “((except for the mean, for which we use a window of size δ_1))”.

Comment 13

“End of para 1, page 10: Please add some more detail to help the reader identify what method is referred to by ” and outlined in Section 2.1 to ascertain ...”

We have added further detail, writing “[...] the method proposed by Dakos et al. (2012) based on ARMA null models [...]”.

Comment 14

“Section 2.5 (page 10) Fig 5a, top panel: When discussing the coefficient of variation say that it’s trend is opposite to the other indicators and hence is shown to perform well”

Because the strong performance in the coefficient of variation was an artefact of our previous analysis approach (see above), the description of this section has changed slightly, addressing the reviewer’s comment.

Comment 15

“Section 3 - Discussion, Rephrase: We analyzed the performance of early warning indicators using the area under the curve to quantify classification performance and using the same methodology with which we analyzed the empirical data, the true positive rate ...”

We have adjusted this sentence accordingly.